# Estimates of the global burden of Japanese encephalitis and the impact of vaccination from 2000-2015

Tran Minh Quan[1,2], Tran Thi Nhu Thao[1,3], Nguyen Manh Duy[1], Tran Minh Nhat[1], Hannah Clapham[1,4,5†]*

[1]Oxford University Clinical Research Unit, Wellcome Trust Asia Program, Ho Chi Minh City, Viet Nam; [2]Biological Science Department, University of Notre Dame, Notre Dame, United States; [3]Virology Department, Institute of Virology and Immunology, University of Bern, Bern, Switzerland; [4]Centre for Tropical Medicine and Global Health, Nuffield Department of Medicine, University of Oxford, Oxford, United Kingdom; [5]Saw Swee Hock School of Public Health, National University of Singapore, Singapore, Singapore

**Abstract** Japanese encephalitis (JE) is a mosquito-borne disease, known for its high mortality and disability rate among symptomatic cases. Many effective vaccines are available for JE, and the use of a recently developed and inexpensive vaccine, SA 14-14-2, has been increasing over the recent years particularly with Gavi support. Estimates of the local burden and the past impact of vaccination are therefore increasingly needed, but difficult due to the limitations of JE surveillance. In this study, we implemented a mathematical modelling method (catalytic model) combined with age-stratifed case data from our systematic review which can overcome some of these limitations. We estimate in 2015 JEV infections caused 100,308 JE cases (95% CI: 61,720–157,522) and 25,125 deaths (95% CI: 14,550–46,031) globally, and that between 2000 and 2015 307,774 JE cases (95% CI: 167,442–509,583) were averted due to vaccination globally. Our results highlight areas that could have the greatest benefit from starting vaccination or from scaling up existing programs and will be of use to support local and international policymakers in making vaccine allocation decisions.

*For correspondence:
hannah.clapham@nus.edu.sg

**Present address:** †Saw Swee Hock School of Public Health, National University of Singapore, Singapore, Singapore

**Competing interests:** The authors declare that no competing interests exist.

## Introduction

Japanese encephalitis (JE) is caused by Japanese encephalitis virus (JEV) – an arbovirus that belongs to the flavivirus genus, family flaviviridae. The main vectors are mosquitoes of the *Culex* genus, especially *Culex tritaeniorhynchus*. These mosquitoes thrive in rice-paddy fields (*Buescher and Scherer, 1959*; *Self et al., 1973*). JEV has a wide range of vertebrate hosts, noticeably the amplifying hosts are thought to be pigs and wading birds (*SAGE Working Group on Japanese encephalitis vaccines, 2014*). Humans are dead-end hosts as viremia is not believed to reach levels that are infectious to mosquitoes (*SAGE Working Group on Japanese encephalitis vaccines, 2014*). Only 1 in 25 to 1 in 1000 infections result in symptoms (*Vaughn and Hoke, 1992*; *SAGE Working Group on Japanese encephalitis vaccines, 2014*). However, the mortality rate of symptomatic cases is high - around 20–30% (*Fischer et al., 2008*), and around 30–50% of survivors experience significant neurological and psychiatric sequelae (*Fischer et al., 2008*).

The first JE case was documented in Japan in 1871 (*WHO, 2015*). In 1924, a first JE outbreak in Japan caused more than 6, 000 cases and 3, 000 deaths in 6 weeks (*Solomon, 2006*). Several outbreaks occurred subsequently in Asia (*Hullinghorst, 1951*; *Erlanger et al., 2009*; *Barzaga, 1990*). More recently, in 2005 large outbreaks occurred in northern India and Nepal, with 5000 cases and

1300 deaths (*Solomon, 2006*). Currently, 24 Asia-Pacific countries are thought to be endemic for JE, with 3 billion individuals at risk of infection (*WHO, 2015*).

The first vaccine was an inactivated mouse brain vaccine produced in Japan, used worldwide for 50 years. Although vaccine production halted in 2006, similar inactivated mouse brain vaccines are still produced locally in South Korea, Taiwan, Thailand and Vietnam (*Yun and Lee, 2014*). The use of the next vaccine, an inactivated a Vero cell vaccine (*SAGE Working Group on Japanese encephalitis vaccines, 2014*), has been gradually replaced (since 1988) by a live attenuated vaccine (SA 14-14-2) produced in China, with PATH support. SA 14-14-2 is now widely used in Asia and funded by Gavi which has led to a great increase in vaccination. This vaccine requires only a single dose, is cheap to produce, and is safer than the mouse brain vaccine (*SAGE Working Group on Japanese encephalitis vaccines, 2014*). In addition, a live attenuated chimeric vaccine was first licensed in Australia in 2012 (*SAGE Working Group on Japanese encephalitis vaccines, 2014*).

WHO recommends two JE surveillance systems that are important for monitoring burdens of JE and changes over time (*WHO, 2019*), (i) a subnational system with sentinel hospitals, or (ii) case-based nationwide surveillance. Each country implements one of these systems depending on available resources (*Hills et al., 2009*). WHO recommends diagnosis using JEV-specific IgM antibody-capture enzyme-linked immunosorbent assay (MAC-ELISA) in CSF at two time points (*Donadeu et al., 2009*; *Burke and Leake, 1988*). Serum samples can be used, but false positives may result from cross-reactivity with other flaviviruses or vaccination (*Solomon et al., 1998*; *Hills et al., 2009*). Other tests that can confirm JE are plaque reduction neutralizing (PRNT), hae-magglutination inhibition (HI), immunohistochemistry or immunofluorescence assay, reverse transcription polymerase chain reaction (RT-PCR) or virus isolation (*Hills et al., 2009*), though these are not often used.

The previous estimate of annual global JE cases was 67,900 with 13,600–20,400 deaths (*Campbell et al., 2011*). For this estimate a systematic review in 2011 collated case incidence data from endemic JE countries. Countries were then stratified into 10 incidence groups (Group A, B, C1-2 and D-I) based on geographic, ecological and vaccine program similarities. The systematic review resulted in 12 key studies, which were then used to infer the incidence rate (IR) of the 10 incidence groups. However the estimation had some limitations; the surveillance quality of the 12 key studies varied, and as the case incidence rate combines both the infection rate and vaccination, (e.g. a low risk of infection with no vaccination could have a similar incidence as a high risk of infection but with high vaccination coverage), it is not possible to estimate the impact of vaccination.

The use of age-stratified case data to infer the FOI has been of use recently for dengue (*Imai et al., 2016*; *Cattarino et al., 2020*; *Rodriguez-Barraquer et al., 2019*). The advantage of this method is that the age distribution will be insensitive to differential reporting or tests used in different places, and the important information from the age distribution remains; the higher the rate of infection, the earlier in life individuals will acquire infection. By fitting models of the infection process (including acquisition of immunity) to this age-stratified data we can quantify this rate of acquiring infection, known as the force of infection (FOI) (*Hens et al., 2010*).

Poor clinical outcomes and lack of specific treatment makes JE prevention a priority. Vaccination is the most effective method of prevention, however it is difficult to decide where vaccination should be implemented or to estimate the quantitative impact of vaccination (*Fischer et al., 2008*). In Nepal, one study estimated 3,011 JE cases were prevented in vaccinated districts from 2006 to 2012 (*Upreti et al., 2017*). Another study in Sarawak Malaysia estimated a 61% reduction in JE cases after the vaccination program, where climate effects were not taken into account, and 45% when the effects of climate were included (*Impoinvil et al., 2013*). The methods used in both these papers require good surveillance data before and after vaccination, which, though data are improving, are currently not widely available. Hence, new approaches are needed to estimate burden and vaccine impact.

In this study, we provide updated global JE burden and vaccination impact estimates using a modelling method which helps overcome some of the limitations of sparse and variable surveillance data. In addition, by simulating the model with and without the undertaken vaccination programs we are able to estimate the impact of vaccination on the number of global JE cases to date and identify areas that would benefit most from future vaccination.

## Results

There are two main stages to our analysis, summarized in flowcharts in *Figure 1*. In the fstage I, we conducted a systematic review to collate age-stratified case data and a literature review to obtain vaccination information. We then fit a model to this data to estimate the transmission intensity or force of infection (FOI) for each study. In stage II, we extrapolated the FOI from our previous estimates to all endemic areas. Using the processed population and vaccination data in all endemic areas, we used the model to generate burden quantities (cases) in two scenarios, with or without the JE vaccination programs that have been implemented.

### Systematic review

A systematic review on October 11[th] 2017 yielded 2337 initial results (*Figure 2*). 407 relevant studies were obtained after eliminating 1931 irrelevant titles and abstracts that were about molecular biology, policy, entomology, hosts other than humans, or were review papers. The obtained studies mainly comprised of reports of JE surveillance or epidemiological studies in one specific location. We also included modelling, economic evaluation or vaccine program assessment studies for possible eligible data sources in the references. We retrieved and read 261 full-text papers. Most of the papers that we could not access were either old or not in English. In the systematic review process, a further four eligible studies were retrieved from references. 202 papers were then excluded as they did not contain age-stratified case data, and another 14 papers were also excluded because they had limited samples (less than 15 cases) or the study's catchment area was not clear. Another four datasets from JE national reports were collated from Taiwan, Japan, and Sri Lanka. Finally, we had 53 studies that contained age-stratified case data (*Figure 2*). 42 of the 53 studies (79%) contained data from after 2000 only, 7 from before 2000 only and three from both time periods (*Figure 2—source data 1*). 34 studies (64%) had data from 1 to 4 year time periods, six studies had data

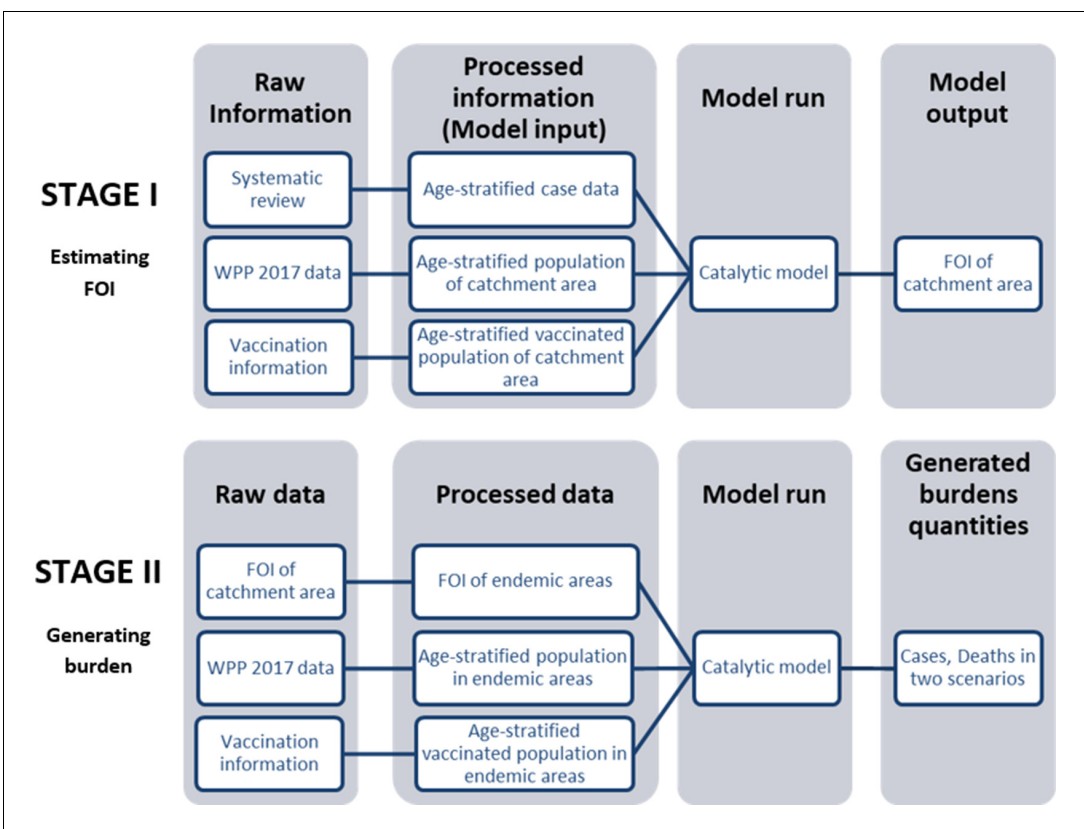

**Figure 1.** Flowchart describes two main stages in our analysis: Estimating FOI (force of infection) and generating burden. In Stage I we estimate FOI (force of infection) of all studies' catchment area. In Stage II we then used the FOI estimates to generate global burden. Abbreviation: WPP: World Population Prospects.

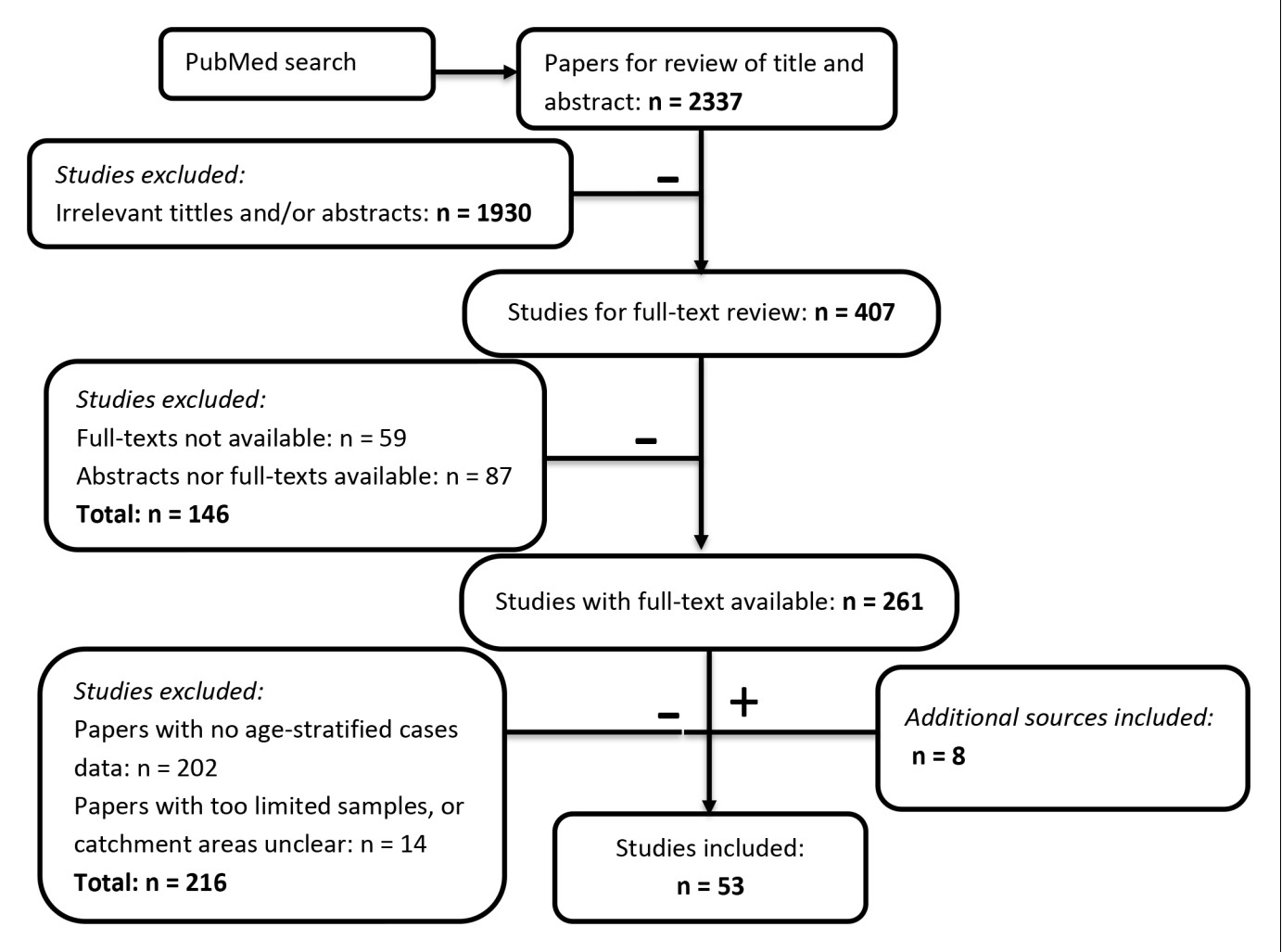

**Figure 2.** Flowchart describing the systematic review procedure searching for Japanese encephalitis age- stratified case data.

The online version of this article includes the following source data and figure supplement(s) for figure 2:

**Source data 1.** Studies from the systematic review that contain age-stratified case data.
**Source data 2.** PRISMA Checklist.
**Figure supplement 1.** PRISMA 2009 flowchart.

for periods of between 5 and 9 years, and 11 studies had data for more than 10 years. The majority of the studies used the WHO JE case definition: JE IgM antibody in CSF or serum as confirmed by MAC-ELISA on patients with acute encephalitis syndrome. In the majority of studies patients were recruited from a sentinel hospital surveillance system, though these ranged in size from one to several hospitals. For studies with a consistent catchment area but for which data was collected in multiple years, we aggregated the age-stratified case data across years. Further details of the selected studies and data, including about catchment areas, sample collection methods, and vaccination programs are in *Figure 2—source data 1*.

We obtained the vaccination information from three main sources: literature review, WHO, and Gavi (*Figure 3—source data 1*). Campaign vaccination information was mainly from Gavi and routine vaccination was from WHO, while the literature contains both. When there were disagreements between the different vaccination information sources, we chose to use the information from the literature review. The total vaccinated population in each country from 2000 to 2015 using information obtained from this data is shown in *Figure 3* and *Figure 3—source data 1*. This information was included as a prior in the model fitting (see *Figure 4—source data 3*).

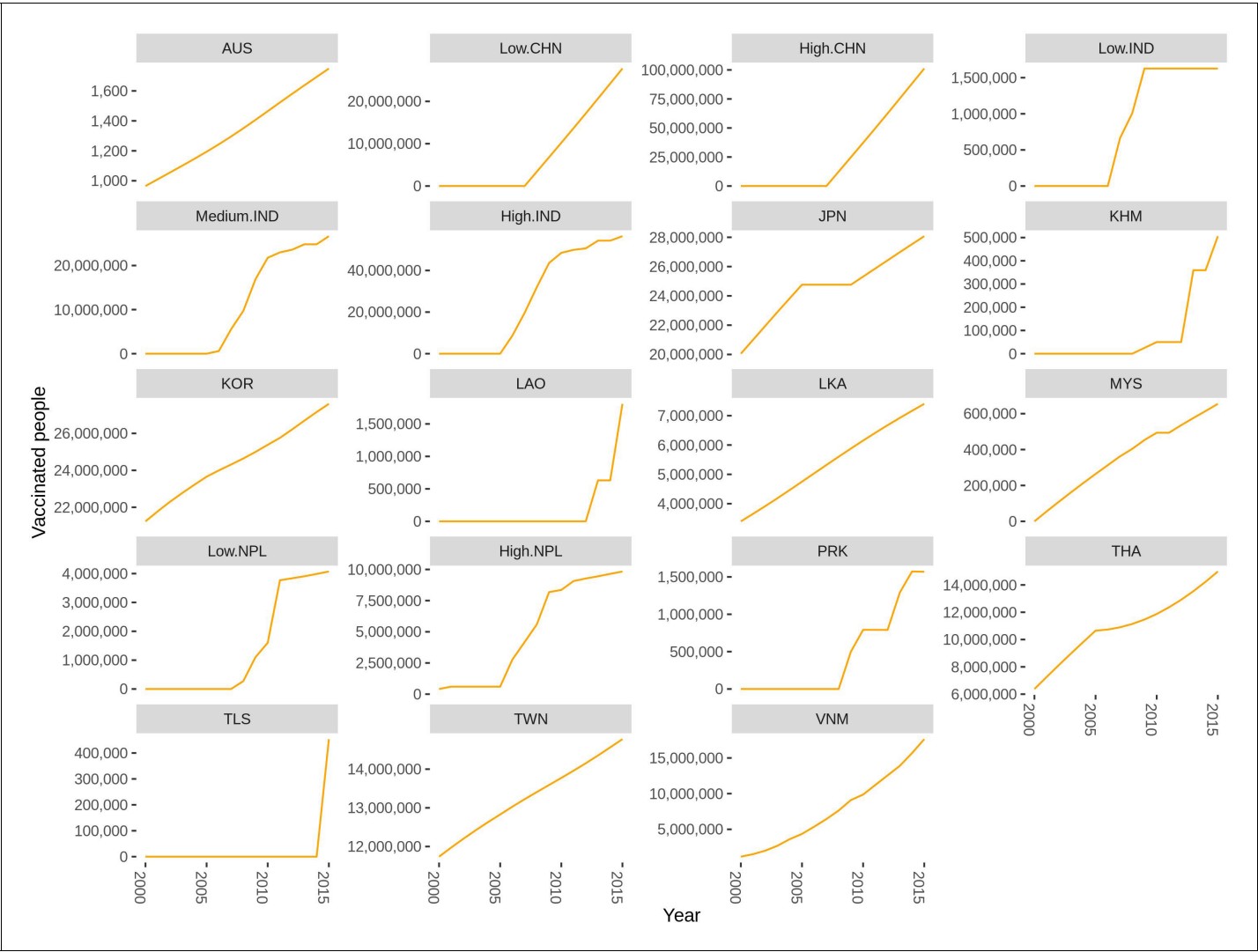

**Figure 3.** Reported number of individuals vaccinated in each region from multiple data sources by region from 2000 to 2015. If the country is not listed there is no vaccination reported. Abbreviations: AUS: Australia, CHN: China, IND: India, JPN: Japan, KHM: Cambodia, KOR: South Korea, LAO: Laos, LKA: Sri Lanka, MYS: Malaysia, NPL: Nepal, PRK: People's Republic of Korea, THA: Thailand, TLS: Timor-Leste, TWN: Taiwan, VNM: Vietnam. The supplementary file: *Figure 3—source data 1* lists the vaccination data and the sources for each country.

The online version of this article includes the following source data for figure 3:

**Source data 1.** Vaccine information and how it was used in our model.

## Force of infection (FOI) estimation from collated age-stratified data

From 53 studies, we made FOI estimates using the catalytic model from 53 unique catchment areas in 15 countries (*Figure 4*). Force of infection (FOI) is the per capita rate at which susceptible individuals are infected by an infectious disease and a catalytic model estimates the FOI from age- stratified case data (*Hens et al., 2010*). All the catalytic models converged well (see convergence plots in *Figure 4—source data 4*) and fit well to data in all but one study (*Figure 4—source data 2* - 95% CIS of model output and case data shown). Our FOI estimates varied from 0.001 (95% CI: 0.000 - 0.002) in Japan to 0.507 (95% CI 0.419 - 0.582) in Guigang in China. Besides those extreme values, FOI were generally between 0.05 and 0.2, with a median of 0.09 (*Figure 4*). In this model-fitting the reporting rate is the proportion of all infections that are reported. The reporting rate includes both the proportion that are symptomatic and the proportion of those cases that present at each hospital or be counted in each surveillance system, so it is the proportion of infections reported. We

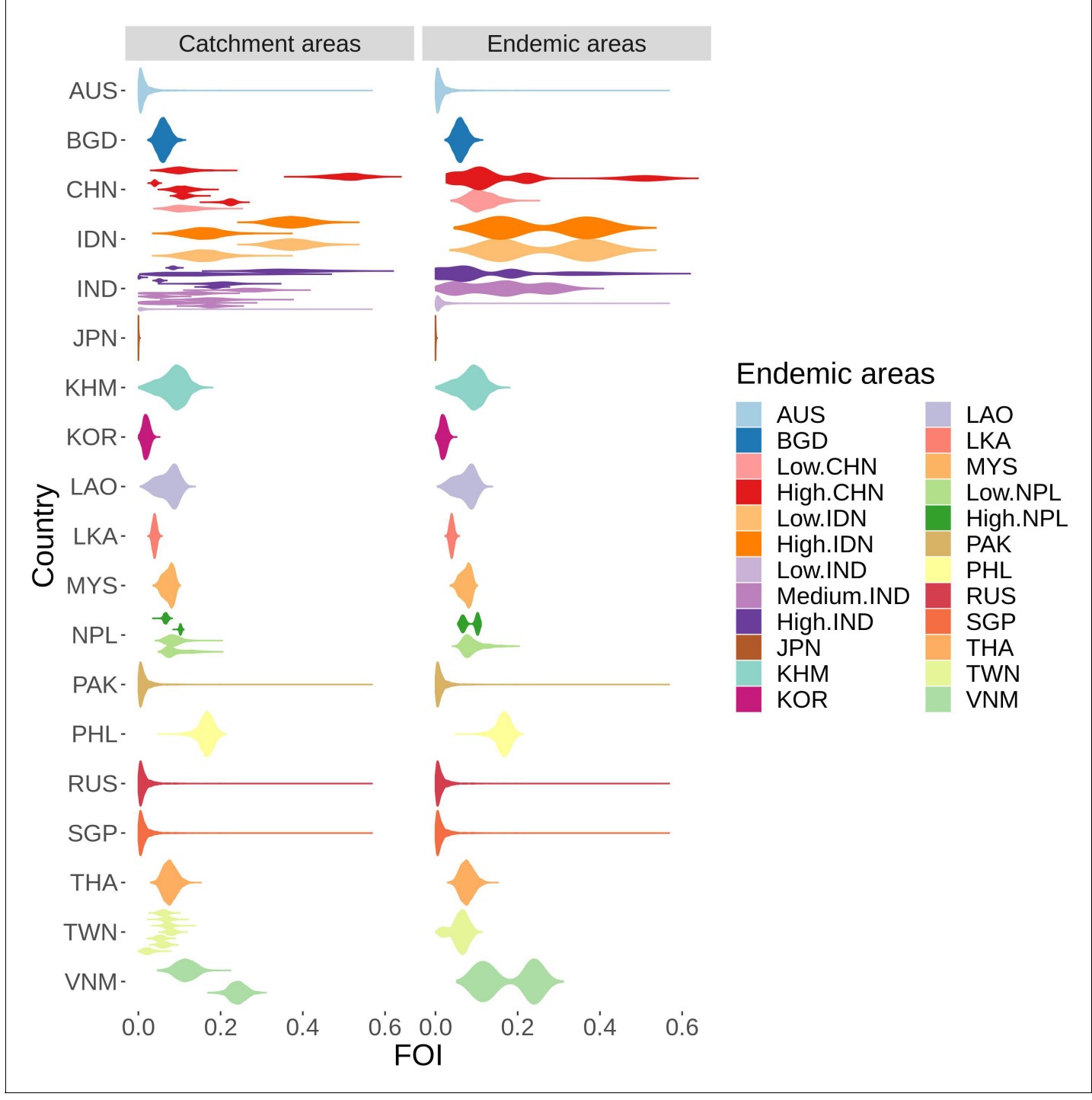

**Figure 4.** FOI distribution estimated from all studies' catchment areas (on the left), each distribution represents FOI from one study, which were used to infer the FOI distribution in all endemic areas (on the right). The colors are coded after the endemic areas as in the legend. Abbreviation: AUS: Australia, BGD: Bangladesh, CHN: China, IDN: Indonesia, IND: India, JPN: Japan, KHM: Cambodia, KOR: South Korea, LAO: Laos, LKA: Sri Lanka, MYS: Malaysia, NPL: Nepal, PHL: Philippines, RUS: Russia, SGP: Singapore, THA: Thailand, TWN: Taiwan, VNM: Vietnam. Countries have low, medium or high following the classification in *Campbell et al., 2011*.

The online version of this article includes the following source data and figure supplement(s) for figure 4:

**Source data 1.** Estimated FOI and studies used/assumptions of 30 endemic areas.
**Source data 2.** Model fit of all age-stratified case data.
**Source data 3.** Susceptible proportion after vaccination in study population.
**Source data 4.** Trace plots for all models fits.

*Figure 4 continued on next page*

*Figure 4 continued*

**Source data 5.** Acf plots for all model fits.
**Figure supplement 1.** Estimated reporting rate from all studies.
**Figure supplement 2.** prior distributions for lambda and rho.

therefore observed a wide variation in estimated reporting rates $\rho$ between studies (*Figure 4—figure supplement 1*). This number is not used in the estimates of cases in the next section. For China, India, Japan, and Nepal, the posterior estimates of the proportion of the population in study $k$ and age group $i$ that remained susceptible after vaccination $s_{k,i}$, for some age groups was slightly different to the prior population vaccination that was included in the model fitting (*Figure 4—source data 3*). When the posterior and prior did not agree, for most datasets this suggested missing vaccination data with the prior saying higher susceptibility than the posterior, however for some areas in India the reverse was estimated.

## Inference of force of infection for all endemic areas

Based on the rules in the methods, we are able to infer FOI from available data for 24 endemic areas (*Figure 4—source data 1*, and *Figure 4*) from across the Campbell et al. groupings. In the Campbell et al. grouping, FOI is assumed to be homogeneous across each country, except Indonesia, China and Nepal with low and high groups, and India with a low, medium and high transmission groups. We kept this grouping for our work except for Indonesia, as the collated data for Indonesia was combined across various provinces across both the low and high incidence areas, we assumed the FOI to be the same in both areas. There were no studies from countries in the *Campbell et al., 2011* group B (Australia, Pakistan, North Korea, Russia, Singapore and the low incidence region in India). Since this group contains extremely low incidence areas, the FOI was assumed to have a lognormal distribution $ln(X) \sim N(0.01, 1)$ (see *Figure 4*).

## Burden and vaccine impact estimation

We estimate the burden from 2000 as the majority of studies used to estimate FOI were from after this time period. We estimate that from 2000 to 2015, there were 1,976,238 (95% CIs: 1,722,533–2,725,647) JE cases globally. By including known annual vaccination information in the catalytic model we estimate that in the same period had there been no vaccination there would have been 2,284,012 (95% CIs: 1,495,964–3,102,542) JE cases. Therefore we estimate that vaccination programs have prevented 307,774 JE cases globally (95% CI: 167,442–509,583) from 2000 to 2015 and vaccination programs similarly prevented 74,769 deaths from JE (95% CIs: 37,837–129,028). We estimate the greatest impact of vaccination from 2005 to 2010 due to large increase in vaccination in China in this time, and the impact of vaccination became more obvious over time (*Figure 5*). In 2015, we estimate vaccination reduced the number of cases globally by around 45,000 (from 145,542 (95% CI: 96,667–195,639) to 100,308 (95% CI: 61,720–157,522) (*Figure 5*).

We estimated the highest number of cases in the high endemic area of China (around 40,000 annual cases in the no vaccination scenario and around 20,000 annual cases in vaccination scenario), and medium or high endemic areas in India (around 20,000 annual cases in no vaccination scenario and 15,000 annual cases in vaccination scenario for each area in recent years). On the contrary, areas like Australia, Brunei, Bhutan and Russia were estimated to have less than 100 annual cases with or without vaccination (*Figure 5*, *Figure 6*). All visualized burden estimates for every years and areas can be found in our interactive map (*Duy, 2018*).

Vaccination impact can be observed in 19 areas where vaccination has been used (*Figure 5*). In areas like the low and high endemic area in China, medium and high endemic area in India, Cambodia, Laos, Nepal, North Korea, and Timor-Leste though vaccination started recently, we estimate that the programs have achieved significant cases averted. Indeed, in the high endemic area in China, the routine vaccination programs only started in 2008 but contributed the most to the global cases reduction, with around 20,000 cases averted in China in 2015. We also observed a clear difference in cases between vaccination and no vaccination scenario in areas with intensive vaccination programs such as South Korea, Sri Lanka, Thailand, Taiwan, and Vietnam. For Japan, Australia, and Malaysia, though vaccination began a long time ago, we estimated there has been minimal vaccine

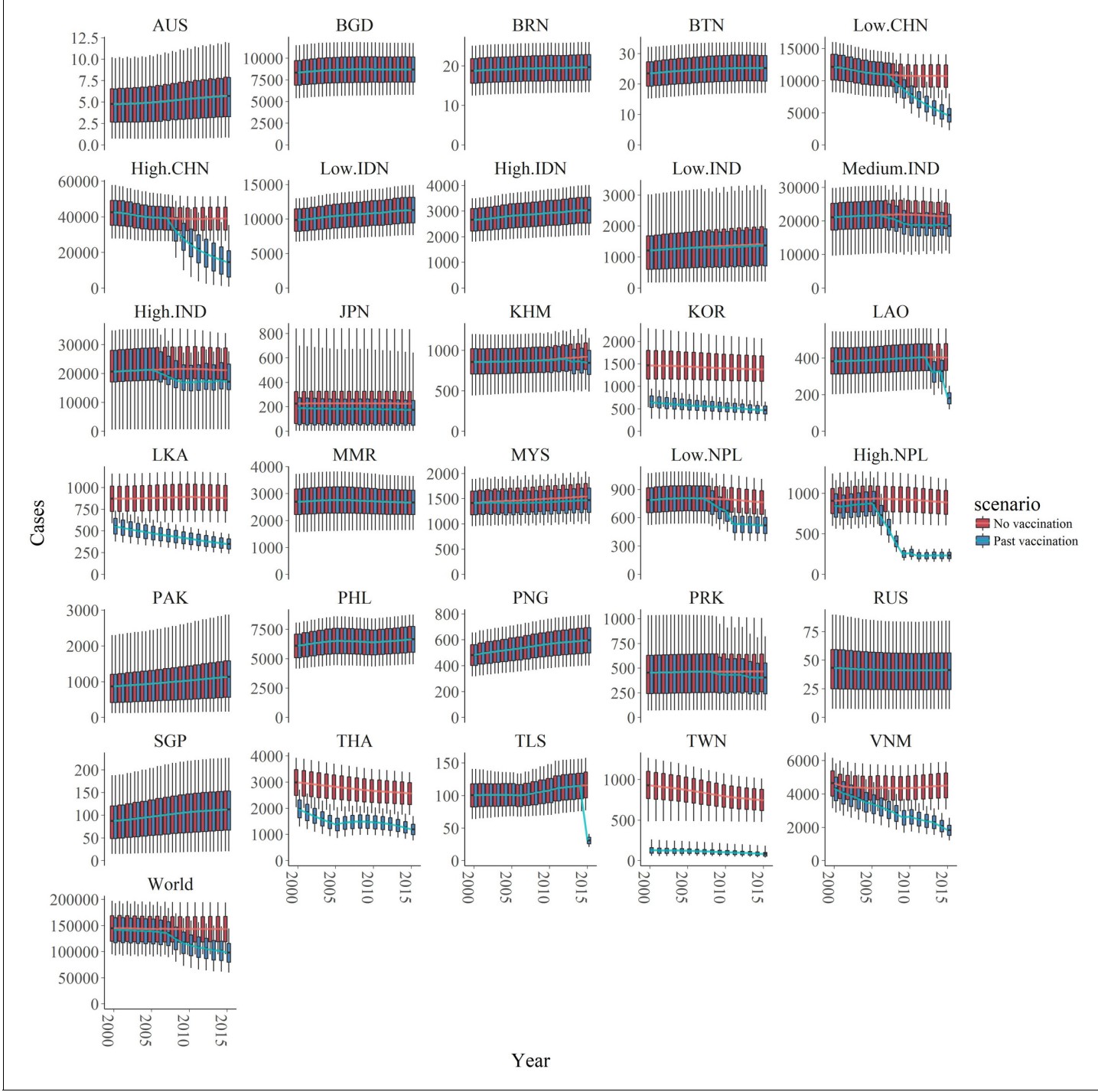

**Figure 5.** Number of estimated cases with and without vaccination of the 30 endemic areas and of the world from 2000 to 2015. The two scenarios, with or without vaccination, are also shown in blue and red respectively. In all areas, the boxplots represent the estimated cases with 95% credible intervals (also shown 1 st quartile, 3rd quartile) with the solid lines showing the mean value of each interval. Abbreviation: AUS: Australia, BGD: Bangladesh, BRN: Brunei, BTN: Bhutan, CHN: China, IDN: Indonesia, IND: India, JPN: Japan, KHM: Cambodia, KOR: South Korea, LAO: Laos, LKA: Sri Lanka, MMR: Myanmar, MYS: Malaysia, NPL: Nepal, PAK: Pakistan, PHL: Philippines, PNG: Papua New Guinea, PRK: North Korea, RUS: Russia, SGP: Singapore, THA: Thailand, TLS: Timor-Leste, TWN: Taiwan, VNM: Vietnam.

The online version of this article includes the following source data for figure 5:

**Source data 1.** Results of sensitivity analyses.

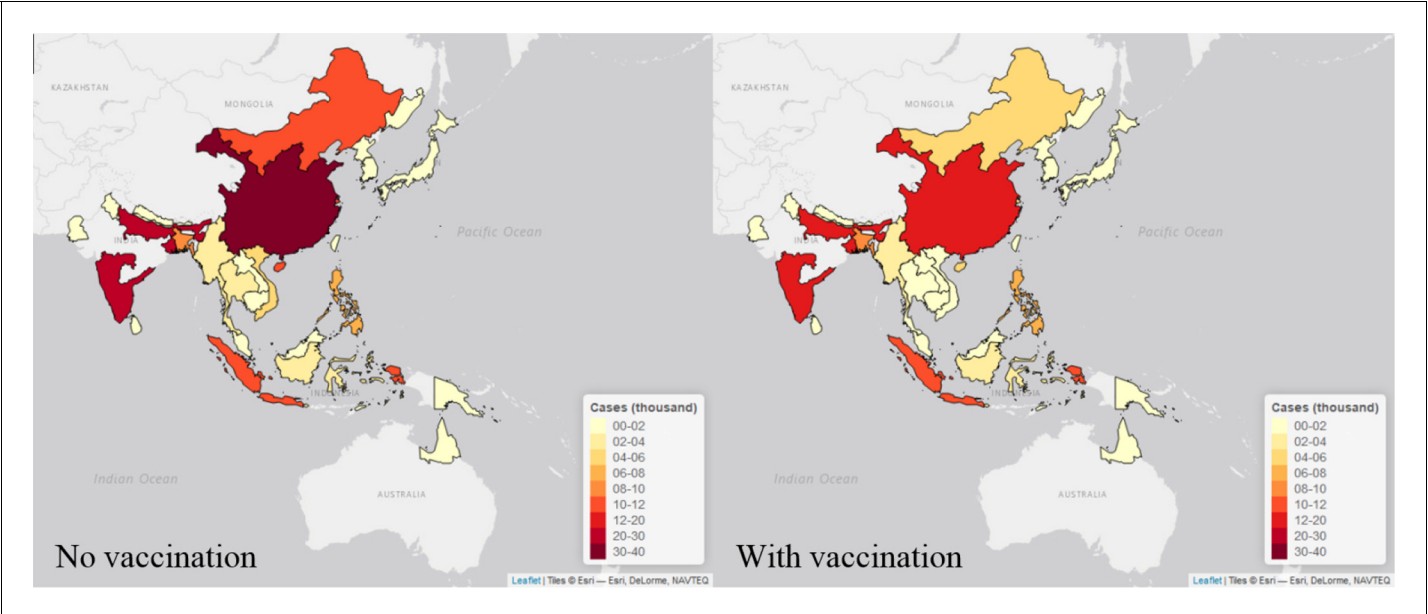

**Figure 6.** Maps of estimated cases (in thousand) in 30 endemic areas for two scenarios in 2015. Each endemic area is shaded in proportion to the area's estimated cases in thousand as seen in the legend, with yellow shade is the lowest value and red shade is the highest value. The map on the left is the estimates from no vaccination scenario, and the right is from the vaccination scenario. The maps were made by *leaflet* package in R (*Joe et al., 2017*).

impact. From the data we collated, no vaccine programs had occurred in in Bangladesh, Brunei, Bhutan, the low and high endemic areas in Indonesia, Myanmar, Pakistan, Philippines, Papua New Guinea, Russia, or Singapore so we estimated of course vaccination has had no impact.

## Sensitivity analysis

To assess the impact of uncertainties in our data and assumptions we performed extensive sensitivity analyses. Sensitivity analyses were conducted for endemic areas with uncertain vaccination coverage data, where both national and subnational data were available (China, India and Nepal), or where we did not have any studies. The majority of the results showed minimal changes compared to our original estimates (*Figure 5—source data 1*). Cases estimated from Taiwan subnational data were higher by about 200 to 400 cases before 2004 (*Figure 5—source data 1*). In some areas, we observed significant differences in the estimated cases when the vaccination coverage was changed: when the vaccination coverage reduced by 10% and 30% in Sri Lanka or by 30% in Thailand and Taiwan, the mean values of estimated cases increase by around 40, 100, 300, and 220 respectively (*Figure 5—source data 1*). However these changes account for a small fraction of our original global estimates. Sensitivity analysis varying the assumed 100% vaccine effectiveness to 90% and 70% showed global case estimates changed minimally with this assumption (*Figure 5—source data 1*). In addition due to concerns about possible changes in FOI over times, we also tested our assumption of constant FOI by fitting multiple-year data to a time-dependent catalytic model. Overall, the annual FOI estimates are comparable with the constant FOI (*Figure 5—source data 1*).

## Discussion

In this paper, we updated the JE burden estimates with a mathematical modelling method using data we collated from a systematic review. We estimated that in 2015 there were around 100,000 JE cases globally. In addition, we estimate that vaccination programs averted around 45,000 JE cases in 2015.

For JE, since humans are dead-end hosts and therefore vaccination does not lead to herd immunity, the FOI we estimate represents the constant spread of the disease from the animal reservoirs to humans. This spread depends on epidemiological factors related to JE transmission such as climate, rural-urban, mosquito distribution (especially *Culex* tritaeniorhynchus), and pig and rice field

distributions (*Le Flohic et al., 2013*). This explains why our estimated FOI varies widely. Looking crudely at the pig density (*Nicolas and Gilbert, 2010*) and a *Culex* tritaeniorhynchus probability maps of *Miller et al., 2012* and *Longbottom et al., 2017* there appears to be a broad correlation of these factors with our estimates. The high FOI estimated in the south of China, Vietnam, and Philippines is consistent with the high pig density and high probability of *Culex* in these areas (for Vietnam and Philippines with *Miller et al., 2012* only). We also estimated high FOI in India and Indonesia; however these countries only have high probability of *Culex* (in *Miller et al., 2012* but not Longbottom et al.) but low pig density in *Longbottom et al., 2017*. This suggests that other potential animal reservoirs may contribute to the transmission in these countries, likely the wading bird or even poultry, although current evidence is limited (*Lord et al., 2015*). In Taiwan and South Korea the current estimated FOI is lower compared to other areas, respectively 0.061 (95% CI 0.013–0.093), and 0.041 (95% CI 0.026–0.057) despite these areas having high probability of *Culex* mosquito and high pig density. These countries have had high JE burdens in the last 40 years, but we do not estimate so for 2000–2015. This could be due to lack of recent data, or perhaps suggests urbanization, which reduces the proximity of humans to pig farms and rice fields (where the mosquitoes thrive), may play an important role in lowering transmission. This could also be due to uncertainties in the long term vaccination information in these areas. Further work will use environmental covariates to gain estimates of FOI on a smaller spatial scale and over time. In addition, changes in these covariates into the future should be considered in estimates of the future vaccine impact.

A strength of our Bayesian approach was the possibility to include prior information on vaccination, but also assess whether this was consistent with the ages distribution of observed cases. For China and Japan we estimated lower susceptible proportions after vaccination in certain age groups compared to calculated proportions from the available data. This suggests that there are a large number of immunized people in certain age groups due to past vaccination, for which we did not have information. In Nepal and India, we also observed differences between the data and estimated susceptible proportion after vaccination, though the vaccination information for these countries was more readily available. There was still an impact of vaccination- but expected impact on the age distribution in the model fitting was not as extreme as the data we collated would have suggested. For India, this artefact as picked up by the model is consistent with data on vaccine efficacy and vaccination coverage data from India. From 2006 to 2011, SA 14-14-2 vaccine was used in India for campaigns. Though the vaccine reported nearly 100% efficacy in vaccine trials and case-control studies (*Kumar et al., 2009*; *Bista et al., 2001*), the efficacy in India was reported to be as low as 30% to 40% (*Vashishtha and Ramachandran, 2015*; *Tandale et al., 2018*) and lower seroconversion has also been reported in India (*Singh et al., 2015*). A previous evaluation of vaccination coverage also showed that the vaccination coverage data in India was lower than reported (*Murhekar et al., 2017*). Further studies are needed to explore whether there are different vaccine efficacies in different places, particularly India, and to explore possible explanations for this. One possible explanation could be cross-reactive immunity to other flaviviruses, or differences in circulating JE genotypes.

Using the FOI from 30 endemic areas, we projected the regional and global JE burdens as well as the vaccine impact. By region, our burdens estimates are highest in China and India, which aligns with previous literature (*Heffelfinger et al., 2017*). Our global estimate of around 100,000 cases annually is about 1.5 times higher than the previous estimate of around 70,000 cases (*Campbell et al., 2011*). Similar patterns are seen for the comparison area by area, in which our case number estimates are either higher than or comparable to the previous estimates (*Table 1*). It is not surprising that our estimates are higher, since our method more robustly takes into account underreporting and different surveillance quality. In addition, the numbers we reported here are time-dependent and not static because our estimates include population changes and the progression of vaccination programs over time.

Though our methods are more robust, collating 53 studies (an additional 41 from the studies used in the previous burden estimate) (*Campbell et al., 2011*), and using age-stratified data to circumvent issues with reporting variation, there are still some limitations. As in the previous estimates of JE burden (*Campbell et al., 2011*), we made inferences for the whole country based on data from a few studies. However in our method we sampled from the FOI estimates from all studies to account for some of this uncertainty and variation. In addition, as in previous studies, a limitation is that we inferred the incidence metric (in our case, FOI) for areas without data, from FOI from other areas, based on previous classification of transmission in these countries. However, our sensitivity

**Table 1.** Comparing annual case estimates from Campbell et al. to our estimates for the year 2015 (as this was the year of estimation of the previous estimates).

Group A: Taiwan, Japan, South Korea; Group B: Australia, low endemic area in India, Pakistan, Russia, Singapore; Group C1: high endemic area in China; Group C2: low endemic area in China; Group D: Cambodia, high endemic area in Indonesia, Laos, Sabah and Labuan in Malaysia, Myanmar, Philippines, Timor-Leste; Group E: low endemic area in Indonesia, Peninsular Malaysia, Papua New Guinea; Group F: high endemic area in India, high endemic area in Nepal; Group G: Bangladesh, Bhutan, Brunei, low endemic area in Nepal; Group H: Medium endemic area in India, Sarawak in Malaysia, Sri Lanka, Thailand, Vietnam; Group I: North Korea.

| Incidence Group | Case numbers: Previous estimates | Case numbers: Our no vaccination scenario Mean estimates (and 95% Cis) | Case numbers: Our vaccination scenario Mean estimates (and 95% Cis) |
|---|---|---|---|
| A | 6 | 2,307 (1,175–3,497) | 863 (453–1,469) |
| B | 2 | 2595 (388-6,243) | 2540 (381-6,071) |
| C1 | 33,849 | 38,789 (26,128–51,482)* | 22,013 (3,778–42,375)* |
| C2 | 28 | 10,752 (7,297–14,152) | 7,094 (4,230–10,579) |
| D | 7917 | 13,710 (9,333–18,135) | 13,700 (9,325–18,125) |
| E | 3645 | 12,932 (8,804–17,059) | 12,932 (8,804–17,059) |
| F | 12,350 | 22,514 (1,503–36,423)* | 17,304 (846-27,930)* |
| G | 1358 | 9,538 (6,322–12,881) | 9,277 (6,133–12,548) |
| H | 8072 | 29,942 (17,431–40,933) | 23,201 (13,647–31,542) |
| I | 670 | 465 (77–1,022)* | 433 (74–912)* |
| *Total* | *67,897* | *143,545 (94,469 – 194,940)* | *109,358 (65,968–156,669)* |

*Our estimates are comparable to the previous estimates.

analysis shows that this does not alter the global burden estimates greatly, though it may affect the country-specific burden estimates. (*Campbell et al., 2011*). Our future work incorporating the epidemiological factors into machine learning algorithms to extrapolate the FOI on smaller spatial scales will help in refining these estimates in the future. Similarly, we assume transmission is constant over time. Further work fitting models with time varying forces of infection as well as looking at covariates of infection that are changing over time, will be necessary for future refinements of these estimates. An additional limitation of our analysis is the uncertainty in the proportion of infections that lead to disease, we sample from this range, and this uncertainty is included in our uncertainty analysis. Further studies, for example in cohorts may enable better estimates of this proportion. In addition, we assumed the FOI is constant across age, only susceptibility changes due to acquistion of immunity, further assessment of seroprevalence studies may be able to assess this further. In addition, though our method accounts for reporting rates within these studies, future work should assess the impact of cross-reactivity and further issues with diagnosis on the estimates, such as including AES data in the fitting. In addition, we did not include papers not in English in our literature review. The papers we could not include were all from China and so further work including these papers should enable better estimation of the FOI in China.

We estimated only the impact of vaccination on cases from 2000 to 2015. Because the impact of vaccination will continue into the future as vaccinated individuals remain protected, our estimate will be an underestimate of the total impact of vaccination. In addition, our estimates will be an underestimate of total vaccine impact as in some places vaccination programs have been running before 2000, and so vaccination had a large impact before 2000. However there is limited information in order to estimate transmission intensity before this time, so we focused our work on 2000–2015. In this paper, we focused on cases (and to some extent deaths) from JE. However because a large number of cases have long-term sequelae after JE infection, focus just on case numbers does not describe fully the total burden of JE. Future work will refine the estimates of the proportion of individuals that die and that experience different long-term sequelae, to generate update our model to estimate JE Disability-Adjusted Life Year (DALY), particularly relevant for use in cost-effectiveness analyses for introduction of vaccination into new locations.

Since JE vaccination does not produce herd immunity, the transmission intensity can only be reduced by influencing the animal transmission cycle. Previous attempts to break the transmission cycle have been vector control and vaccination in pigs and wading birds, and this has been considered in modelling work (*Khan et al., 2014*). However they were either ineffective or up to now have been deemed economically and logistically intensive (*Fischer et al., 2008*). Further work considering pig vaccination in the context of these updated estimates of the burden of JE should be considered. We estimate that despite not interrupting transmission, human vaccination can be an effective strategy to reduce JE case numbers. This can be seen from the estimate that the majority of the reduction in global burden is due to the routine vaccination program in China from 2008. We estimate that India, Timor-Leste, and Vietnam also have high transmission intensity, and residual cases despite vaccination, and therefore could further benefit from scaling-up the existing vaccination program (*Figure 3—source data 1*). We estimated high transmission intensity in Indonesia, Papua New Guinea, and Philippines where there are no current vaccination programs, suggesting that vaccination in these areas should be a future priority. Future smaller scale estimates will support decisions on where within these countries could be best targeted for vaccination. For areas with a long history of JE vaccination (see *Figure 3—source data 1*). such as South Korea, Sri Lanka, Thailand and Taiwan, (*Figure 4*), we estimate a substantial vaccine impact (*Figure 5*), though with cases still occurring. In other countries with a long vaccination history however, we estimate a minimal impact of vaccination (*Figures 4* and *5*), due to low estimated transmission intensity in Japan, low vaccination coverage in Malaysia, or both in Australia (though age-stratified data were not available in Australia). Our estimate of transmission intensity for Japan also has great uncertainty, as half the studies included data pre-2000 and we were able to find limited information on the long-running vaccination program there. In addition, there are limitations to how well our method will work, given the high rate of vaccination there. This may mean we are under-estimating the impact of vaccination in Japan. Further work with serological data both from humans and animals and further exploration of the drivers of JE transmission will help refine this estimate.

Assessing JE disease burden and vaccination program performance is important though difficult due of the lack solid surveillance programs worldwide. In our paper, we are able to estimate the disease burden and vaccine impact using a modelling method that is able to overcome some of the limitations of current surveillance. We estimate annually there are still 100,000 cases of JE in Asia, making a 2/3 of all cases of this severe but vaccine preventable still not being averted. The majority of remaining cases are focussed in countries with still developing healthcare systems. Given there is a cheap vaccination now available, our results will help with the rational assessment of JE vaccination cost and benefit for each country and will help guide Gavi and other international and national public health agencies in making decision on their future investment into JE vaccination.

# Materials and methods

## Systematic review

We performed a systematic review to find all available age-stratified case data for Japanese encephalitis in PubMed. We used the search terms 'epidemiology' or 'incidence' or 'prevalence' or 'public health' or 'surveillance' or 'distribution' in all fields with 'Japanese encephalitis' in the title or abstract. All titles and abstracts were screened and we selected those in which the study contained age-stratified case data. We retrieved the full-texts for these selected abstracts and the abstracts were read by two independent reviewers to extract the age-stratified case data. From each study we also collected other information about the catchment areas, sample collection methods, diagnosis tests, and regional vaccination programs from the papers. A final consensus was reached for the final list of eligible full-texts. If abstracts were not available, the two independent individuals also tried to access and examine the full-texts. We also searched online for age-stratified case data from national JE surveillance reports.

We obtained vaccination information either from the study itself or from the literature review. Based on the review of JE vaccination programs reported from the World Health Organization (WHO) (*Heffelfinger et al., 2017*), we found that previous vaccination programs had occurred in 13 countries. We then undertook a literature search to find all vaccination information (target age group, vaccination coverage, types of vaccine used, years of vaccination) for these countries. We

also collated historical routine vaccination program from country reported administrative doses data time series (from 2000 to 2015) compiled from WHO-UNICEF Joint Reporting (*World Heath Organization, 2018*) and additional data from Gavi.

## Force of infection estimation

Force of infection (FOI) is the per capita rate at which susceptible individuals are infected by an infectious disease. In this study, we used a basic Muench's catalytic model (*Muench, 1958*) to estimate the constant age and time independent FOI using the case data we extracted during the systematic review process. A similar approach has been used to estimate the global dengue transmission intensity (*Imai et al., 2016*; *Rodriguez-Barraquer et al., 2019*). As humans are dead-end hosts for JE, the FOI represents the FOI from the animal reservoir, and therefore is not impacted by human vaccination. This means vaccination can be included in the model simply as a removal of susceptible individuals by vaccination (or a reduction in risk of infection in this vaccinated group depending on vaccine efficacy) and will not alter the FOI. Therefore in this model, individuals can become immune to infection either by natural infection (depending on the force of infection) or vaccination.

To estimate the FOI (notated as $\lambda_k$), for each study $k$, taking into account vaccination and reporting rate for each study $k$, the modelled number of cases in a specific age group $i$ is:

$E_{k,i} = P_{k,i}pop_{k,i}s_{k,i}\rho_k$, where

$$P_{k,i} = \left( e| \; |-\lambda_k a_{k,i}^l - e^{-\lambda_k \left( a_{k,i}^u + 1 \right)} \right) \tag{1}$$

Where $P_{k,i}$ estimates the incident rate of infection in each age group $i$ (with lower and upper $a_{k,i}^l$ and $a_{k,i}^u$ respectively), accounting for force of infection and susceptibility in that age group due to natural infection before this age. $pop_{k,i}$ is the population size in each age group $i$ of each study $k$, calculated from World Population Prospects 2017 data (*United Nations-Department of Economic and Social Affairs-Population Division, 2017*). $s_{k,i}$ is the estimated susceptible proportion in each age group $i$ after vaccination for population in study $k$. The prior distribution of $\lambda_k$ was an uninformative non-negative, normal distribution, $\lambda_k \sim Normal(0, 1000)$. To include the uncertainty in the vaccination information, we used an informative prior: $s_{k,i} Beta\left( \Phi\left(1 - s_{k,i}'\right), \Phi s_{k,i}' \right)$, with $s_{k,i}'$ is the proportion of the population that remain susceptible after vaccination in age group $i$ of study $k$, calculated from the vaccination information and the population demographics in the study's catchment area. $\Phi$ represents the uncertainty of the vaccination information (we set $\Phi = 5$ to account for the possibility that this information was incomplete or did not reflect the actual vaccinations delivered. The chosen uncertainty value represents moderate trust in the vaccination information. The value of 7 or 10 gave a very strong belief, hence not chosen here (these analyses are not shown)) (see *Figure 4—source data 2* for priors). $\rho_k$ is the reporting rate for each study, which is comprised of symptomatic rate and the reporting rate of the surveillance system and accounts for the different surveillance qualities of the different studies. Since $\rho_k$ contains the symptomatic rate which reported to be less than 1% (*SAGE Working Group on Japanese encephalitis vaccines, 2014*; *Vaughn and Hoke, 1992*), we used an informative prior: $\rho_k \sim Beta(0.1, 9.9)$.

The log-likelihood function for each study $k$ is the sum of the multinomial log-likelihood and Poisson log-likelihood of total cases across all age groups.

$$L_k^{MN+P} = log(t_k!) - \sum_i log(C|\;|k,i!) + \sum_i C_{k,i}log\left(\frac{E_{k,i}}{\sum_i E_{k,i}}\right) + t_k log\left(\sum_i E_{k,i}\right) - \sum_i E_{k,i} - log(t_k!) \tag{2}$$

Where $t_k$ is the total number of cases and $C_{k,i}$ is the number of age-stratified cases in age group $i$ in each study $k$. $E_{k,i}$ is the modelled number of cases in a specific age group $i$.

For each dataset, we fit the model in a Bayesian framework in RStan (*Stan Development Team, 2016*), estimating parameters $\lambda_k, \rho_k, s_{k,i}$. RStan uses a No-U-Turn sampler (NUTS) (*Hoffman and Gelman, 2014*), a variant of Hamiltonian Monte Carlo to obtain posterior simulation (*Stan Development Team, 2016*). The parameters $s_{k,i}, \rho_k$ were all estimated on a logit scale. We started 4 random chains, each with 16000 iterations and 50% burn-in period. Smaller step size of the

Hamiltonian transition was manually set by increasing the adapt delta parameter in RStan to be 0.99. Model convergence was assessed visually.

We assumed that the JE vaccine has 100% effectiveness, which is reasonable given the reported high effectiveness of the vaccine (*World Health Organization, 2012a*; *WHO, 2014*; *World Health Organization, 2012b*) and that the protection acquired from natural infection or vaccination was life-long.

For our estimate, the endemic areas were defined to be the same as in the previous JE burden estimate (*Campbell et al., 2011*). For China, India, Nepal and Indonesia, where transmission intensity is diverse these countries were broken down to low, medium, or high endemic areas. In total, there are 30 endemic areas, spanning 24 countries. We inferred the FOI for each endemic area based on the FOI estimated from collated studies. The inference was based on two rules: (1) For each area, the FOI was obtained by sampling from the estimated FOI of all the studies that had catchment areas within that endemic area (if any). (2) For endemic areas in which no studies were conducted, the FOI was inferred to be equal to the FOI of the area in the same incidence group defined by *Campbell et al., 2011*.

## Burden and vaccine impact estimation

Once the distributions of inferred FOIs for each endemic area were obtained, we generated the distributions of the estimates of the number of cases in each year $t$ (from 2000 to 2015) in endemic area $d$ for each age group $a$ from 0 to 99 years old and scenario $m$ (described below) using the function (similar to the model used to estimate FOI (*Equation 1*)):

$$cases_{m,d}(a,t) = \left(1 - e^{-\lambda_d}\right)e^{-\lambda_d a}\rho_{sym}pop_{m,d}(a,t) \qquad (3)$$

$\lambda_d$ is the FOI of that area (assumed constant over time and age independent) which is sampled from the posterior estimates from the previous model fitting. The term $e^{-\lambda_d a}$ is the decrease in proportion of susceptible population due to natural infection. $\rho_{sym}$ is symptomatic rate, sampled from $Uniform\left(\frac{1}{500}, \frac{1}{250}\right)$ (*SAGE Working Group on Japanese encephalitis vaccines, 2014*). The symptomatic rate is the proportion of infections that are estimated to show symptoms, this is different to the reporting rate in the FOI estimation section, which includes differential reporting and testing by study. $pop_{m,d}(a,t)$ is the susceptible population of age $a$ in endemic area $d$ in year $t$ under scenarios $m$ and was interpolated from World Population Prospects 2017 data (*United Nations-Department of Economic and Social Affairs-Population Division, 2017*). To assess the impact of previous vaccination programs, the population $pop_{m,d}(a,t)$ was different for each vaccination scenario $m$: with or without vaccination. The vaccination scenario used the collated information about past vaccination programs and assumed that the number of vaccinations given each year to each age meant that this number of the relevant age groups in the population were not susceptible to infection from this year onwards. This takes into account aging of the vaccinated population and any changes in the vaccination programs over time.

Although the mortality rate of JE varies, the reported ranges are from 20-30% (*Fischer et al., 2008*). We sampled the mortality rate from $Uniform(0.2, 0.3)$ and multiplied it by the estimated number of $cases_{m,d}(a,t)$ to generate age-specific JE-induced deaths.

All code and data are available at: https://github.com/tranquanc123/JE_burden_estimates (*Quan, 2020*; copy archived at https://github.com/elifesciences-publications/JE_burden_estimates).

## Additional information

### Funding

| Funder | Grant reference number | Author |
| --- | --- | --- |
| Bill and Melinda Gates Foundation and Gavi | Vaccine Impact Modelling Consortium | Tran Minh Quan Tran Thi Nhu Thao Nguyen Manh Duy Tran Minh Nhat |
| Wellcome Trust | 089276/B/09/7 | Hannah Clapham |

The funders had no role in study design, data collection and interpretation, or the decision to submit the work for publication.

**Author contributions**
Tran Minh Quan, Resources, Data curation, Software, Formal analysis, Investigation, Visualization, Methodology, Writing - original draft; Tran Thi Nhu Thao, Resources, Data curation, Writing - review and editing; Nguyen Manh Duy, Data curation, Software, Formal analysis, Writing - review and editing; Tran Minh Nhat, Resources, Data curation, Undertook the literature review, Reviewed and approved the final manuscript for submission; Hannah Clapham, Conceptualization, Supervision, Methodology, Writing - original draft, Writing - review and editing

**Author ORCIDs**
Tran Minh Quan (iD) https://orcid.org/0000-0003-3337-161X
Tran Minh Nhat (iD) https://orcid.org/0000-0002-9500-8341
Hannah Clapham (iD) https://orcid.org/0000-0002-2531-161X

**Decision letter and Author response**
Decision letter https://doi.org/10.7554/eLife.51027.sa1
Author response https://doi.org/10.7554/eLife.51027.sa2

## Additional files

**Supplementary files**
• Transparent reporting form

**Data availability**

This study conducted a literature review and collated all data on age-stratified JE cases from these papers. The full list of these papers and the data extracted is available in the supplement. The code and data is available at https://github.com/tranquanc123/JE_burden_estimates (copy archived at https://github.com/elifesciences-publications/JE_burden_estimates).

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
