## [Decision Letter]

**Acceptance summary:**

Significance statement: Because of its high morbidity and mortality, Japanese Encephalitis (JE) is an important public health concern in Asia. Vaccination is a realistic preventive strategy but requires reliable epidemiologic surveillance across endemic countries. To overcome this problem, the authors used age-stratified data combined with mathematical modelling to estimate the global transmission intensity and burden of JE, as well as the impact of vaccination, thus expanding on earlier work. The annual number of cases is about 50% higher than originally estimated, correcting for under-reporting and variations in countries' surveillance systems.

**Decision letter after peer review:**

Thank you for submitting your article "Estimates of the global burden of Japanese Encephalitis and the impact of vaccination from 2000-2015" for consideration by *eLife*. Your article has been reviewed by three peer reviewers, and the evaluation has been overseen by Mark Jit as Reviewing Editor and Eduardo Franco as the Senior Editor. The reviewers have opted to remain anonymous.

The reviewers have discussed the reviews with one another and the Reviewing Editor has drafted this decision to help you prepare a revised submission.

This review and modelling study updates global estimates of the burden of disease due to Japanese Encephalitis (JE). Based on a systematic review of the current literature, the paper makes revised models of the disease burden of JE and of the effects of vaccination. The revised estimate of the global burden of JE is 100,000 cases per year. This is an entirely plausible number, and, importantly, the paper estimates that only about 1/3 of cases are prevented by vaccination.

This is an important paper in the field, and should be published, with some modifications. The estimates are important given the lack of robust surveillance systems. The goal of this study – to improve understanding of the global burden of JE and the impact of existing vaccination efforts – is critical for informing future public health strategies. The manuscript is clear and well-written.

General comments

1) Could you add more detail about the Bayesian model used to estimate the force of infection and other key parameters as currently there is not sufficient information to review it? It would be useful to see both prior and posterior (including joint posterior) distributions for all the parameters, as well as trace plots, autocorrelation plots and any other diagnostic plots that you think are relevant. We may ask a Bayesian statistician to review this once we have received it.

2) How were asymptomatic infection incorporated in the estimations? Were individuals with asymptomatic infection considered susceptible or immune after infection? Particularly since the rate of asymptomatic to symptomatic JE is wide ranging (25:1 to 1000:1), this may have implications on estimations.

3) The model makes a number of assumptions which may have modified the accuracy of their conclusions. These assumptions include (i) vaccine efficacy is 100%, (ii) force of infection remains constant over time, (iii) the relative incidence of JE across different countries has remained the same since the Campbell et all review, (iv) population distribution is homogenous and (v) the force of infection is the same for all age groups.

Assumptions (i) and (ii) have been directly addressed by sensitivity analyses, and do not seem to affect the model outcomes that much, despite the fact that neither is true. The argument made that the force of infection need not vary over time to make the model valid is a little difficult to follow from Figure 5—figure supplement 1M for the non-modeller reader. See below for more on vaccine efficacy in India. Assumption (iii) appears not to make very much difference as most of the areas for which there are little data are areas of low incidence. However it is worth noting that the assumption itself otherwise may be incorrect – future workers would be wrong to take this as a precedent. Assumptions (iv) and (v) have not been addressed by the authors. They too may not affect the conclusions that much but they should at least be discussed.

Specific comments

1) Results section. Reliability of JE case data. While MAC-ELISA is the WHO recommended procedure for JEV diagnosis, concerns have been raised regarding the sensitivity (50-70%) and specificity of this test (E.g., Robinson et al., 2010 Am J Trop Med Hyg, Dubot-Peres et al., 2015, Lancet Infect Dis). Further to this, parallel testing for dengue-specific antibodies is often not done. It is not clear from Figure 2—figure supplement 1 whether they reported testing samples against other endemic antigenically related flaviviruses (e.g. dengue). Given that the diagnostic test employed by most studies included in your analysis has a low confirmatory level for JEV infection, could this uncertainty somehow be incorporated in your study (also the possibility that diagnostic trends have changed over the study time period)? Additionally, could any adjustments be made where diagnostic methods less confirmatory than MAC-ELISA were used? At the very least you should acknowledge this issue and discuss how it may have impacted their results and conclusions.

2) Subsection “Force of infection estimation from collated age-stratified data”: FOI estimates are very low for Japan. Could this just be because there's successful vaccination making the data uncertain due to low numbers? E.g. there are some papers that suggest the FOI in parts of Japan can be much higher, e.g. 10%, or 100 fold higher. What would the effect of this be on the model? See for example Konishi et al. Vaccine 2002; 21:98-107 and Vaccine 2006; 24:3054-6.

3) “We estimated that the proportion of the population in study 𝑘 and age group 𝑖 that remained susceptible after vaccination 𝑠𝑘,𝑖 , was different to the prior collated vaccination information in areas such as China, India, Japan, and Nepal (Figure 2—figure supplement 4).” -This sentence is hard to understand without the figure – what/where is Figure 2—figure supplement 4?

4) Subsection “Inference of force of infection for all endemic areas” – Presumably incidence group B refers to the same incidence group as per Campbell et al., 2011? This isn't actually explained until several pages later – it would be much easier for a reader who is not so well versed in the JE literature to understand if this were to be defined earlier.

5) Subsection “Burden and Vaccine Impact estimation” – only approx. 15% of JE cases have been prevented! This figure is worthy of more discussion, as although it has approximately doubled in recent years, still only about 1/3 of cases are prevented. Clearly there is still much to do.

6) Subsection “Burden and Vaccine Impact estimation”/Figure 5 – Are you suggesting that there has been no effect of JE vaccination in India? See also Discussion paragraph three.

7) Discussion paragraph three – Efficacy was high in case control studies too, not only in trials. See for example Ohrr et al. Lancet 2005; 366: 1375-78 and Kumar et al.,2009.

8) This efficacy estimate is now formally published and would make a better reference (Tandale et al., 2018). Vaccine efficacy does appear to be lower in India, for reasons which are not entirely clear. A lower rate of sero-conversion to the SA14-14-2 vaccine has also been described in India (Singh et al., 2015). Have you provided further support for these observations? Can you suggest why the efficacy of SA14-14-2 vaccine should be lower in India?

9) Table 1 – State more clearly that these are case numbers, and what the numbers in brackets are.

10) Discussion paragraph six – The paper acknowledges that its estimates will be underestimates for various reasons. However one important reason has been left out – that is that JE can be difficult to diagnose and there is a miss rate whereby even genuine cases will test negative because the test has been done too early in the illness (it takes a week for CSF IgM to be positive in close to 100% of cases and lumbar punctures are often not repeated this late). The authors should they acknowledge that these reported figures will be minimum estimates. Eg see Susan Hills' work in Nepal showing all AES went down after JE vaccination, not only JE (Upreti et al. Am. J. Trop. Med. Hyg., 88(3), 2013, pp. 464-468).

11) Could there be a table of countries with JE vaccination programs?

12) Subsection “Force of infection estimation”: Limited JE surveillance systems could lead to inaccurate reporting of disease. The authors took this into consideration in their estimation of number of cases in a specific age group by including pk (reporting rate) into consideration. It's not clear how the reporting rate accounts for different surveillance qualities. For example, in countries where JE virus IgM antibody was not confirmed with CSF, how was surveillance quality (or potential heterogeneity) addressed?

13) The paper notes that convergence was assessed visually. Were you also able to assess statistical measures of goodness of fit, such as R2 or residual plots to confirm the visual assessments?

14) Figure 3 plots the "estimated number of vaccinated individuals by region". Please make it clear in the figure caption that these are not model-based estimates, but calculated from vaccination data. Even referring to these values as "Reported number of individuals vaccinated in each region, summed [?] from multiple data sources". If any adjustments were made in making these calculations, please specify.

15) Figure 3 – Please provide further information on the data from Gavi and WHO. Does the Gavi data represent the number of doses delivered or health records of vaccinated individuals or follow-up survey data? Is the WHO joint reporting data publicly available? If so, please include the link. What uncertainties exist in these various different data sources and why are these not incorporated in Figure 3? The point estimates imply a high level of confidence in these values. Please consider acknowledging uncertainty in Figure 3.

16) Figure 3 – The paper later acknowledges uncertainty in these values and include it in their analysis through the selection of priors. Why are many of the priors so constrained for some countries e.g. Laos, Bangladesh, Philippines (Figure 4—figure supplement 3)? Does this not suggest a high level of confidence? How much data at each location informed the posterior?

[Editors' note: further revisions were suggested prior to acceptance, as described below.]

Thank you for resubmitting your work entitled "Estimates of the global burden of Japanese Encephalitis and the impact of vaccination from 2000-2015" for further consideration by *eLife*. Your revised article has been evaluated by Eduardo Franco as Senior Editor and Mark Jit as Reviewing Editor.

A second review round for your submission is complete. Four reviewers spent a considerable amount of time examining your work. We are happy to see the effort you made at amending the paper to accommodate the concerns and suggestions from the reviewers of the original version. Once again, we are unable to accept it in its present form for publication. However, we are willing to consider a new revised version if you can address the additional items in the reviewers' critiques. These comments were edited to eliminate redundancy and to help you focus on the revisions. In general, the required revisions have to do with clarity of the presentation and making the model reproducible, which requires providing the data and code in a user-friendly format.

1) Clarity of presentation:

Please revise Figure 3 to use integers, not scientific numbers for ease of interpretation.

Subsection “Force of infection (FOI) estimation from collated age-stratified data”: The authors indicate that the reporting rate = total rate of symptomatic JE (I assume that means with or without presentation to facility/surveillance system), plus % of people presenting to a facility or surveillance system. As written, it seems like this would lead to double counting and result in overinflated estimates? Please explain or clarify the text.

What was the magnitude of variation between prior and posterior estimates of susceptible post vaccination?

Table 1: What is the time frame for these estimates? (annual or for the 2000-2015 timeframe?)

Subsection “Force of infection estimation”: Can you give brief rationale for why you set your uncertainty of vaccination information to 5? The text references Figure 4—figure supplement 3 but this graph is difficult to read and does not include sources for selection of prior distributions.

Supplemental Appendix: In general, this information is important but I find the figures in Figure 4—figure supplement 3 to be a bit impenetrable. I think this could be addressed fairly simply by renaming the graphs (for example, from s_prop[2]), legends, and adding footnotes.

There are a few formatting issues – in particular some versions of some of the figures in the PDF file are of very poor resolution.

This may be a function of the publisher's format for review, but I found the figures, legends and nomenclature very hard to follow. I don't fully understand the hierarchy of "Figure X- Supp Y" etc. As long as the authors proofread the typeset manuscript very carefully this will be fine and it is not a reason for re-review – rather a word of caution that in the current form it's difficult to read and this needs to be improved on publication. Labelling each figure on the page on which it is presented would help a lot. My reading of the current figures is that they are appropriate and in some cases improved from the first version. eg Figure 3—source data – I am not sure what/where this is? What/where is S2 Fig?)

Figure 4—figure supplement 4-6 – I agree these should remain supplementary as they are indeed confusing for the non-statistician reader.

For those not familiar with FOI rates, it might help to have an explanation of what the number actually means – eg does FOI 0.05 mean 5% of the population infected per year? This would make it more accessible.

I agree with the authors choice of wording concerning the efficacy in India – I do not have any better explanation than this – except possibly a nutritional effect – but this is speculative and it is not clear why this should be different to other countries.

2) Issues related to the Bayesian model:

A) The overall model written in stan seems good. It is a shame that the data were not provided alongside the R code, and it would seem easy to remedy this (I would recommend packaging them together as an RMarkdown file along with the curated dataset). *eLife* is committed to reproducible research and having easy to re-run code and data is quite important for a study whose conclusions are entirely dependent on the underlying statistical model and available data. Asking people to extract data from a Word document is prone to errors etc.

B) I couldn't work out whether or not there was propagation of uncertainty from model 1 (the FOI model) to model 2 (the burden and vaccine impact model). i.e. is the model 2 using a point estimate from model 1, or is it integrating over the full posterior? If there is no propagation of uncertainty I would recommend this is changed (otherwise it will underestimate the uncertainty in the final output); if there is propagation of uncertainty, then please make it clearer.

C) The notation confused me in parts. Is λ_d_ the FOI or is it FOI(λ_d_)? If so what is the function "FOI"? This is probably obvious to people used to dealing with these catalytic models, but as I've never seen one before I had trouble understanding.

D) A prior of N(0,1000) is probably not the best choice. If λ_d_ is the FOI, then Figure 4 shows that all the values are between 0 and ~0.65. So values of 2000 are completely implausible. Your chains will get better mixing if you use weakly informative priors. See https://github.com/stan-dev/stan/wiki/Prior-Choice-Recommendations for a nice overview of recommendations from the stan team (your choice is under the "not usually recommended" category!). A guess of a better prior might be an exponential distribution (values are always positive), with mean value close to what you expect in most cases.

Please provide an easy to use (e.g. RMarkdown) implementation, this would have the data as a csv file, for example, included.

---

## [Author Response]

General comments1) Could you add more detail about the Bayesian model used to estimate the force of infection and other key parameters as currently there is not sufficient information to review it? It would be useful to see both prior and posterior (including joint posterior) distributions for all the parameters, as well as trace plots, autocorrelation plots and any other diagnostic plots that you think are relevant. We may ask a Bayesian statistician to review this once we have received it.

We have added details on the model fitting methods and results throughout the paper and uploaded additional supplementary files as detailed below:

1) In the Materials and methods we added more detail on RStan algorithm: “RStan uses a No-U-Turn sampler (NUTS) (Hoffman and Gelman, 2014), a variant of Hamiltonian Monte Carlo to obtain posterior simulation (Stan Development Team, 2016).”

2) Convergence plots have been uploaded as Figure 4—figure supplement 4 and autocorrelation plots as Figure 4—figure supplement 5.

3) In the Results we added details on the model fit and point the reader towards the convergence plots: “Force of infection (FOI) is the per capita rate at which susceptible individuals are infected by an infectious disease. All the catalytic models converged well (see convergence plots in Figure 4—figure supplement 4 and autocorrelation plots Figure 4—figure supplement 5) and fit well to data in all but one study (Figure 4—figure supplement 1- 95% CIS of model output and case data shown).”

4) The prior and posteriors for the susceptible proportion after vaccination are shown in Figure 4—figure supplement 3. The priors for the reporting rate and λ have been plotted in Figure 4—figure supplement 6. I thought about putting on the results figures, but thought that would be confusing for the non-statistician reader. I can add if you prefer.

5) In our results, we estimated that the posterior estimate of the proportion of the population in study k and age group i that remained susceptible after vaccination s_k,i_, was different to the prior collated population vaccination that was included in the model fitting vaccination information in areas for areas such as China, India, Japan, and Nepal (Figure 4—figure supplement 3).

2) How were asymptomatic infection incorporated in the estimations? Were individuals with asymptomatic infection considered susceptible or immune after infection? Particularly since the rate of asymptomatic to symptomatic JE is wide ranging (25:1 to 1000:1), this may have implications on estimations.

Yes this range is indeed wide. This is a large part of the uncertainty in our estimates as we sample from across this range. We have added emphasis to this in the Discussion: “An additional limitation of our analysis is the uncertainty in the proportion of infections that lead to disease, we sample from this range, and this uncertainty is included in our uncertainty analysis. Further studies, for example in cohorts may enable better estimates of this proportion.”

3) The model makes a number of assumptions which may have modified the accuracy of their conclusions. These assumptions include (i) vaccine efficacy is 100%, (ii) force of infection remains constant over time, (iii) the relative incidence of JE across different countries has remained the same since the Campbell et all review, (iv) population distribution is homogenous and (v) the force of infection is the same for all age groups.Assumptions (i) and (ii) have been directly addressed by sensitivity analyses, and do not seem to affect the model outcomes that much, despite the fact that neither is true. The argument made that the force of infection need not vary over time to make the model valid is a little difficult to follow from Figure 5—figure supplement 1M for the non-modeller reader. See below for more on vaccine efficacy in India. Assumption (iii) appears not to make very much difference as most of the areas for which there are little data are areas of low incidence. However it is worth noting that the assumption itself otherwise may be incorrect – future workers would be wrong to take this as a precedent. Assumptions (iv) and (v) have not been addressed by the authors. They too may not affect the conclusions that much but they should at least be discussed.

We thank the reviewers for their careful thoughts about our assumptions. We have added more discussion of these assumptions into the Discussion section.

For assumption (iii) we have added additional statement about this in the Discussion: “Similarly, we assume transmission is constant over time. Further work fitting models with time varying forces of infection as well as looking at covariates of infection that are changing over time, will be necessary for future refinements of these estimates.”

For assumption (iv) population age distribution is assumed homogenous within a country, but not across age- the age distribution is as per population data for each country. I think the way this is currently stated is not clear and unnecessary here, so have deleted this sentence. The source of the population age distribution for each country is given: “pop_k,i_ is the population size in each age group i of each study k, calculated from World Population Prospects 2017 data (United Nations-Department of Economic and Social Affairs-Population Division, 2017).”

For assumption (v), we have added the following to the Discussion: “In addition, we assumed the FOI is constant across age, only susceptibility changes, further assessment of seroprevalence studies may be able to assess this further.” The vaccine in India comments are addressed below.

Specific comments1) Results section. Reliability of JE case data. While MAC-ELISA is the WHO recommended procedure for JEV diagnosis, concerns have been raised regarding the sensitivity (50-70%) and specificity of this test (E.g., Robinson et al., 2010 Am J Trop Med Hyg, Dubot-Peres et al., 2015, Lancet Infect Dis). Further to this, parallel testing for dengue-specific antibodies is often not done. It is not clear from Figure 2—figure supplement 1 whether they reported testing samples against other endemic antigenically related flaviviruses (e.g. dengue). Given that the diagnostic test employed by most studies included in your analysis has a low confirmatory level for JEV infection, could this uncertainty somehow be incorporated in your study (also the possibility that diagnostic trends have changed over the study time period)? Additionally, could any adjustments be made where diagnostic methods less confirmatory than MAC-ELISA were used? At the very least you should acknowledge this issue and discuss how it may have impacted their results and conclusions.

This is a really important comment, the sensitivity of the test and how this varies between studies in different places at different times, will be taken into account by the reporting rate- missing cases across all ages will be estimated as a lower reporting rate for this study, but will not alter the FOI estimates (see comment above). These are not the reporting rates used in the burden and vaccine estimates- so it will not alter our case estimates as well. We have added the following to the Discussion: “In addition, though our method accounts for reporting rates within these studies, future work should assess the impact of cross-reactivity and further issues with diagnosis on the estimates, such as including AES data in the fitting.”

2) Subsection “Force of infection estimation from collated age-stratified data”: FOI estimates are very low for Japan. Could this just be because there's successful vaccination making the data uncertain due to low numbers? Eg there are some papers that suggest the FOI in parts of Japan can be much higher, eg 10%, or 100 fold higher. What would the effect of this be on the model? See for example Konishi et al. Vaccine 2002; 21:98-107 and Vaccine 2006; 24:3054-6.

We thank the reviewer for this comment. We have added to the Discussion so the section on the estimates for Japan now reads as follows: “Our estimate of transmission intensity for Japan also has great uncertainty, as half the studies included data pre-2000 and we were able to find limited information on the long-running vaccination program there. In addition, there are limitations to how well our method will work, given the high rate of vaccination there. This may mean we are under-estimating the impact of vaccination in Japan. Further work with serological data both from humans and animals and further exploration of the drivers of JE transmission will help refine this estimate.”

We have removed discussion of Japan from elsewhere in the Discussion.

3) “ We estimated that the proportion of the population in study κ and age group i that remained susceptible after vaccination s_k,i_, was different to the prior collated vaccination information in areas such as China, India, Japan, and Nepal (Figure 2—figure supplement 4).” -This sentence is hard to understand without the figure – what/where is Figure 2—figure supplement 4?

Rephrased as follows: “In our results, we estimated that the proportion of the population in study and age group that remained susceptible after vaccination , was different to the prior population vaccination that was included in the model fitting for areas such as China, India, Japan, and Nepal (Figure 4—figure supplement 3)”. Apologies we gave the wrong figure reference previously- it has been updated to Figure 4—figure supplement 3.

4) Subsection “Inference of force of infection for all endemic areas” – Presumably incidence group B refers to the same incidence group as per Campbell et al., 2011? This isn't actually explained until several pages later – it would be much easier for a reader who is not so well versed in the JE literature to understand if this were to be defined earlier.

Thanks for pointing this out, we have added more detail on the grouping names in the Introduction and then added the reference and reference to the Campbell groupings in the text and Figure 4 legend.

5) Subsection “Burden and Vaccine Impact estimation” – only approx. 15% of JE cases have been prevented! This figure is worthy of more discussion, as although it has approximately doubled in recent years, still only about 1/3 of cases are prevented. Clearly there is still much to do.

We thank the reviewer for focussing on this. We have tried to add emphasis to this by rewording the last section of the manuscript. The final paragraph now reads as: “We estimate annually there are still 100,000 cases of JE in Asia, making a 1/3 of all cases of this severe but vaccine preventable still not being averted. The majority of remaining cases are focussed in countries with still developing healthcare systems. Given there is a cheap vaccination now available, our results will help with the rational assessment of JE vaccination cost and benefit for each country and will help guide Gavi and other international and national public health agencies in making decision on their future investment into JE vaccination.”

6) Subsection “Burden and Vaccine Impact estimation”/Figure 5 – Are you suggesting that there has been no effect of JE vaccination in India? See also Discussion paragraph three.

We apologise this was unclear- there has been little impact in the low incidence areas because the incidence rate is estimated to be very low even without vaccination.

We have clarified that this is only the low incidence sub-area of India. In we state that “We estimated the highest number of cases …….and medium or high endemic areas in India (around 20,000 annual cases in no vaccination scenario and 15,000 annual cases in vaccination scenario for each area in recent years).”

We have added clarification: “There was still an impact of vaccination- but expected impact on the age distribution in the model fitting was not as extreme as the data we collated would have suggested.”

7) Discussion paragraph three – Efficacy was high in case control studies too, not only in trials. See for example Ohrr et al. Lancet 2005; 366: 1375-78 and Kumar et al., 2009.

Thanks- we added the following references: (Kumar, Tripathi, and Rizvi, 2009); Bista et al., 2001),

8) This efficacy estimate is now formally published and would make a better reference (Tandale et al., 2018). Vaccine efficacy does appear to be lower in India, for reasons which are not entirely clear. A lower rate of sero-conversion to the SA14-14-2 vaccine has also been described in India (Singh et al., 2015). Have you provided further support for these observations? Can you suggest why the efficacy of SA14-14-2 vaccine should be lower in India?

Thanks very much for the additional references. This is a really interesting point for further study. We have added the Tandale et al. reference and added Singh reference with the statement: “lower seroconversion has also been reported in India (Singh et al., 2015).”

This section now reads: “Further studies are needed to explore whether there are different vaccine efficacies in different places, particularly India and to explore possible explanations for this. One possible explanation could be cross-reactive immunity to other flaviviruses or differences in circulating JE genotypes.”

Does the reviewer have any thoughts on why this might be?

9) Table 1 – State more clearly that these are case numbers, and what the numbers in brackets are.

Added to table column headers.

10) Discussion paragraph six – The paper acknowledges that its estimates will be underestimates for various reasons. However one important reason has been left out – that is that JE can be difficult to diagnose and there is a miss rate whereby even genuine cases will test negative because the test has been done too early in the illness (it takes a week for CSF IgM to be positive in close to 100% of cases and lumbar punctures are often not repeated this late). The authors should they acknowledge that these reported figures will be minimum estimates. Eg see Susan Hills' work in Nepal showing all AES went down after JE vaccination, not only JE (Upreti et al. Am. J. Trop. Med. Hyg., 88(3), 2013, pp. 464-468).

(Response as for comment 1) This is a really important comment, the sensitivity of the test and how this varies between studies in different places at different times, will be taken into account somewhat by the reporting rate- missing cases across all ages will be estimated as a lower reporting rate for this study, but will not alter the FOI estimates (see comment above). These are not the reporting rates used in the burden and vaccine estimates- so it will not alter our case estimates as well. We have added the following to the Discussion: “In addition, though our method accounts for reporting rates within these studies, future work should assess the impact of cross-reactivity and further issues with diagnosis on the estimates, such as including AES data in the fitting.”

We have also added the following for clarification on the reporting vs symptomatic rate: “The symptomatic rate is the proportion of infections that are estimated to show symptoms, this is different to the reporting rate in the FOI estimation section, which includes differential reporting and testing by study.”

11) Could there be a table of countries with JE vaccination programs?

This is in Figure 3—source data 1which was not properly referenced in the previous version. In order to guide the reader to this we have added reference throughout this section to Figure 3—source data 1.

12) Subsection “Force of infection estimation”: Limited JE surveillance systems could lead to inaccurate reporting of disease. The authors took this into consideration in their estimation of number of cases in a specific age group by including pk (reporting rate) into consideration. It's not clear how the reporting rate accounts for different surveillance qualities. For example, in countries where JE virus IgM antibody was not confirmed with CSF, how was surveillance quality (or potential heterogeneity) addressed?

(Response as for comment 1 and 10). This is a really important comment, the sensitivity of the test and how this varies between studies in different places at different times, will be taken into account somewhat by the reporting rate- missing cases across all ages will be estimated as a lower reporting rate for this study, but will not alter the FOI estimates (see comment above). These are not the reporting rates used in the burden and vaccine estimates- so it will not alter our case estimates as well. We have added the following to the Discussion: “In addition, though our method accounts for reporting rates within these studies, future work should assess the impact of cross-reactivity and further issues with diagnosis on the estimates, such as including AES data in the fitting.”

13) The paper notes that convergence was assessed visually. Were you also able to assess statistical measures of goodness of fit, such as R2 or residual plots to confirm the visual assessments?

We further describe the model fit and point the reader towards the correct file showing the models fits in the Results: “and fit well to data in all but one study (Figure 4—figure supplement 1- 95% CIS of model output and case shown).” and added into the supplementary materials the trace plots for the model fits.

14) Figure 3 plots the "estimated number of vaccinated individuals by region". Please make it clear in the figure caption that these are not model-based estimates, but calculated from vaccination data. Even referring to these values as "Reported number of individuals vaccinated in each region, summed [?] from multiple data sources". If any adjustments were made in making these calculations, please specify.

The legend title has been updated to: “Reported number of individuals vaccinated in each region from multiple data sources by region from 2000-2015. If the country is not listed there is no vaccination reported. ”. We also added a signpost to the supplementary file that contains all the source information: The supplementary file: Figure 3—source data 1 lists the data and the sources for each country.

15) Figure 3 – Please provide further information on the data from Gavi and WHO. Does the Gavi data represent the number of doses delivered or health records of vaccinated individuals or follow-up survey data? Is the WHO joint reporting data publicly available? If so, please include the link. What uncertainties exist in these various different data sources and why are these not incorporated in Figure 3? The point estimates imply a high level of confidence in these values. Please consider acknowledging uncertainty in Figure 3.

All vaccine data from the two sources, interpretation and what was chosen if there are discrepancies is shown in Figure 3—source data 1 and we have added the reference to this in the Figure 3 legend and in the manuscript. Figure 3 legend has also been altered to read “Reported number of individuals vaccinated in each region from multiple data sources. If the country is not listed there is no vaccination reported. ” rather than estimated to make it clear that this is simply the reported numbers.

The data is available in Figure 3—source data 1 and we have added reference to this data throughout this section.

16) Figure 3 – The paper later acknowledges uncertainty in these values and include it in their analysis through the selection of priors. Why are many of the priors so constrained for some countries e.g. Laos, Bangladesh, Philippines (Figure 4—figure supplement 3)? Does this not suggest a high level of confidence? How much data at each location informed the posterior?

We have altered the legend to Figure 3: “Reported number of individuals vaccinated in each region from multiple data sources by region from 2000-2015. If the country is not listed there is no vaccination reported.”

Thanks for looking at this figure so closely. This figure is susceptible proportion after vaccination and so 1 means no vaccination. In places like Laos, Bangladesh, Philippines where there has been no reported vaccination, we assumed in the prior vaccination would have been reported had it happened, thus the tight intervals. We have added clarification to the legend of Figure 4—figure supplement 3 as follows “This is the susceptible proportion so (1- vaccinated proportion) therefore estimates of 1 here means no vaccination.”

In addition we have clarified the section on discussing when the prior and posterior do not match. “In our results, We estimated that the proportion of the population in study k and age group i that remained susceptible after vaccination s_k,i_, was different to the prior collated population vaccination that was included in the model fitting vaccination information in areas such as China, India, Japan, and Nepal (Figure 4—figure supplement 3).”

[Editors' note: further revisions were suggested prior to acceptance, as described below.]

In general, the required revisions have to do with clarity of the presentation and making the model reproducible, which requires providing the data and code in a user-friendly format.1) Clarity of presentation:Please revise Figure 3 to use integers, not scientific numbers for ease of interpretation.

Done

Subsection “Force of infection (FOI) estimation from collated age-stratified data”: The authors indicate that the reporting rate = total rate of symptomatic JE (I assume that means with or without presentation to facility/surveillance system), plus % of people presenting to a facility or surveillance system. As written, it seems like this would lead to double counting and result in overinflated estimates? Please explain or clarify the text.

We apologise this was misleading. We have updated the text to read: “In this model-fitting the reporting rate is the proportion of all infections that are reported, this is both the proportion that are symptomatic, and the proportion of those cases that present at each hospital or be counted in each surveillance system, we do not estimate these separately”

What was the magnitude of variation between prior and posterior estimates of susceptible post vaccination?

We re-ordered this to be clear this was only for some countries and described the variation more: “For China, India, Japan, and Nepal in our results, the posterior estimates of the proportion of the population in study k and age group i that remained susceptible after vaccination s_k,i_, for some age groups was slightly different to the prior population vaccination that was included in the model fitting for areas such as China, India, Japan, and Nepal (Figure 4—figure supplement 3). When the posterior and prior did not agree, for most datasets this suggested missing vaccination data, with the prior saying higher susceptibility than the posterior, however for some areas in India the reverse was estimated.”

Table 1: What is the time frame for these estimates? (annual or for the 2000-2015 timeframe?)

We have updated the figure legend to: Table1: Comparing annual case estimates from Campbell et al. to our estimates for the year 2015 (as this was the year of estimation of the previous estimates).

Subsection “Force of infection estimation”: Can you give brief rationale for why you set your uncertainty of vaccination information to 5?

Added: “to account for the possibility that this information was incomplete or did not reflect the actual vaccinations delivered. The chosen uncertainty value represents moderate trust in the vaccination information. The value of 7 or 10 gave a very strong belief, hence not chosen here (these analyses are not shown))”.

The text references Figure 4—figure supplement 3 but this graph is difficult to read and does not include sources for selection of prior distributions.Supplemental Appendix: In general, this information is important but I find the figures in Figure 4—figure supplement 3 to be a bit impenetrable. I think this could be addressed fairly simply by renaming the graphs (for example, from s_prop[2]), legends, and adding footnotes.

We have redone the Figure 4—figure supplement 3 with the figures being a much larger size and with a description for each study and relabelled the y-axis label.

We have uploaded a better quality figure of Figure 4—figure supplement 1 too also with larger figures.

There are a few formatting issues – in particular some versions of some of the figures in the PDF file are of very poor resolution.

We apologise for this. We have improved the format and re-uploaded. Resolution problem: Figure 3 and 4; Format problem: Figure 4—figure supplement 1 and Figure 4—figure supplement 3 which we have updated with better quality figures

This may be a function of the publisher's format for review, but I found the figures, legends and nomenclature very hard to follow. I don't fully understand the hierarchy of "Figure X- Supp Y" etc. As long as the authors proof read the typeset manuscript very carefully this will be fine and it is not a reason for re-review – rather a word of caution that in the current form it's difficult to read and this needs to be improved on publication. Labelling each figure on the page on which it is presented would help a lot. My reading of the current figures is that they are appropriate and in some cases improved from the first version. eg Figure 3—source data 1 – I am not sure what/where this is? What/where is S2 Fig?)

We agree this was not intuitive, unfortunately *eLife* would not allow uploading of one supplementary information and so supplementary files had to be uploaded in this way. We have checked all references, and where necessary corrected and added more details in the supplementary files to help the reader understand what they are.

Figure 4—figure supplement 4-6 – I agree these should remain supplementary as they are indeed confusing for the non-statistician reader.

Great, we will leave in the supplementary information.

For those not familiar with FOI rates, it might help to have an explanation of what the number actually means – eg does FOI 0.05 mean 5% of the population infected per year? This would make it more accessible.

We are nervous to do this, because the relationship is not linear and we would have to give a number of examples. We hope that the explanation is enough for people to get an intuited understanding of what the FOI means.

I agree with the authors choice of wording concerning the efficacy in India – I do not have any better explanation than this – except possibly a nutritional effect – but this is speculative and it is not clear why this should be different to other countries.

We agree.

2) Issues related to the Bayesian model:A) The overall model written in stan seems good. It is a shame that the data were not provided alongside the R code, and it would seem easy to remedy this (I would recommend packaging them together as an RMarkdown file along with the curated dataset). eLife is committed to reproducible research and having easy to re-run code and data is quite important for a study whose conclusions are entirely dependent on the underlying statistical model and available data. Asking people to extract data from a Word document is prone to errors etc.

Thank you for your suggestion. We apologise this was not done previously. The data and codes are uploaded here https://github.com/tranquanc123/JE_burden_estimates, and we have added the link to the github at the end of the Materials and methods.

B) I couldn't work out whether or not there was propagation of uncertainty from model 1 (the FOI model) to model 2 (the burden and vaccine impact model). I.e. is the model 2 using a point estimate from model 1, or is it integrating over the full posterior? If there is no propagation of uncertainty I would recommend this is changed (otherwise it will underestimate the uncertainty in the final output); if there is propagation of uncertainty, then please make it clearer.

Yes there was propagation of uncertainty of the FOI from model 1 to model 2. We have added the following to the burden and vaccine impact estimation methods section to make this clear: “Once the distributions of inferred FOIs for each endemic area were obtained, we generated the distributions of the estimates of the number of cases “ and “ FOI… which is sampled from the posterior estimates from the previous model fitting.”

C) The notation confused me in parts. Is λ_d_ the FOI or is it FOI(λ_d_)? If so what is the function "FOI"? This is probably obvious to people used to dealing with these catalytic models, but as I've never seen one before I had trouble understanding.

We apologise this was unclear, we have added the following sentence: “FOI (notated as λ_d_)”

D) A prior of N(0,1000) is probably not the best choice. If λ_d_ is the FOI, then Figure 4 shows that all the values are between 0 and ~0.65. So values of 2000 are completely implausible. Your chains will get better mixing if you use weakly informative priors. See https://github.com/stan-dev/stan/wiki/Prior-Choice-Recommendations for a nice overview of recommendations from the stan team (your choice is under the "not usually recommended" category!). A guess of a better prior might be an exponential distribution (values are always positive), with mean value close to what you expect in most cases.

Yes you are right. In hindsight we probably could have run for a shorter time with different prior distribution. The parameter values were on the log scale so the range is not quite as wide (and positive). We used the prior as in a previous paper (10.1093/infdis/jiv470) Though this may speed up our model convergence, and would involve a re-running of the model system, we do not think it would have altered our conclusions, as convergence was reached, so would request to not re-run this now.Next time we will probably use a β distribution or exponential as the reviewer suggests.

Please provide an easy to use (e.g. RMarkdown) implementation, this would have the data as a csv file, for example, included.

All csv data and Rmarkdown are now uploaded at: https://github.com/tranquanc123/JE_burden_estimates. Apologies we did not do this before. This is also referenced at the end of the Materials and methods section.